# Fine Characterization of the Macromolecular Structure of Huainan Coal Using XRD, FTIR, ^13^C-CP/MAS NMR, SEM, and AFM Techniques

**DOI:** 10.3390/molecules25112661

**Published:** 2020-06-08

**Authors:** Dun Wu, Hui Zhang, Guangqing Hu, Wenyong Zhang

**Affiliations:** 1School of Earth and Space Sciences, University of Science and Technology of China, Hefei 230026, China; wudun@ustc.edu.cn (D.W.); huguangqing117@163.com (G.H.); 2Exploration Research Institute, Anhui Provincial Bureau of Coal Geology, Hefei 230088, China; 3Hefei National Laboratory for Physical Sciences at the Microscale (HFNL), University of Science and Technology of China, Hefei 230026, China

**Keywords:** fine characterization, macromolecular structure, XRD, FTIR, ^13^C NMR, Huainan coal

## Abstract

Research on the composition and structure of coal is the most important and complex basic research in the coal chemistry field. Various methods have been used to study the structure of coal from different perspectives. However, due to the complexity of coal and the limitations of research methods, research on the macromolecular structure of coal still lacks systematicness. Huainan coalfield is located in eastern China and is the largest coal production and processing base in the region. In this study, conventional proximate analysis and ultimate analysis, as well as advanced instrumental analysis methods, such as Fourier transform infrared spectroscopy (FITR), X-ray diffraction (XRD), ^13^C-CP/MAS NMR, and other methods (SEM and AFM), were used to analyze the molecular structure of Huainan coal (HNC) and the distribution characteristics of oxygen in different oxygen-containing functional groups (OCFGs) in an in-depth manner. On the basis of SEM observation, it could be concluded that the high-resolution morphology of HNC’s surface contains pores and fractures of different sizes. The loose arrangement pattern of HNC’s molecular structure could be seen from 3D AFM images. The XRD patterns show that the condensation degree of HNC’s aromatic ring is low, and the orientation degree of carbon network lamellae is poor. The calculated ratio of the diameter of aromatic ring lamellae to their stacking height (La/Lc = 1.05) and the effective stacking number of aromatic nuclei (Nave = 7.3) show that the molecular space structure of HNC is a cube formed of seven stacked aromatic lamellae. The FTIR spectra fitting results reveal that the aliphatic chains in HNC’s molecular structure are mainly methyne and methylene. Oxygen is mainly –O–, followed by –C=O, and contains a small amount of –OH, the ratio of which is about 8:1:2. The molar fraction of binding elements has the approximate molecular structure C_100_H_76_O_9_N of organic matter in HNC. The results of the ^13^C NMR experiments show that the form of aromatic carbon atoms in HNC’s structure (the average structural size *X_b_* of aromatic nucleus = 0.16) is mainly naphthalene with a condensation degree of 2, and the rest are aromatic rings composed of benzene rings and heteroatoms. In addition, HNC is relatively rich in ≡CH and –CH_2_– structures.

## 1. Introduction

Coal is not only the most important energy source, but also an important raw material for modern metallurgical and chemical industries [1]. Understanding the molecular structure of coal is the basis of coal development and utilization and coal chemistry research [2], and is also the main research direction of coal chemists today [3,4,5]. The complexity and heterogeneity of coal make it more difficult to study its structure [6,7]. Understanding the macromolecular structure of coal (especially low-rank coal (LRC)) is conducive to the efficient and accurate utilization of coal and is of great significance for extending the coal industry chain. Structural information on coal can be used to understand and control the processing of coal, such as liquefaction [8], pyrolysis [9], combustion [10], and high-performance materials (grapheme [11,12], carbon nanotubes [13,14], and carbon fiber sheets [15,16]).

With the development of basic physics and precision instruments, the identifiable range and accuracy of modern physical analysis instruments have been significantly improved. At present, in the field of coal’s molecular structure, physical instruments, such as X-ray diffraction (XRD), Fourier transform infrared spectrum (FTIR), solid-state nuclear magnetic resonance (NMR), and X-ray photoelectron spectroscopy (XPS), are employed to characterize its chemical structure and analyze its statistical structure. Physical research methods have the advantages of causing little damage to the molecular structure and a high sensitivity for quantitative analysis, and are widely used in the study of coal’s molecular structure.

When studying the crystal structure of a substance by X-ray diffraction analysis, the diffraction direction is related to the shape and size of the unit cell, while the diffraction intensity is related to the arrangement of atoms in the unit cell. Therefore, it can analyze crystals such as graphite well. Coal is not a crystal, but X-ray diffraction analysis can also reveal the ordering of carbon atoms in coal. Since Lisse et al. [17] discovered the X-ray, it has been widely used in the structural analysis of coal. Lewis [18] analyzed the distribution of molecular groups in coal by using the characteristic that electrons scatter X-rays. Based on the experimental results of X-ray diffraction and the Bragg equation, the structural parameters of microcrystals can be calculated, including the diameter (L_a_) of aromatic ring lamellae, the stacking height (L_c_) of aromatic ring lamellae, and the distance (d_oo2_) between aromatic ring lamellae. Song et al.’s [19] research shows that the size of microcrystalline units in coal is mainly controlled by the coal grade, i.e., with the increase of the coal grade, L_a_ increases, d_002_ decreases, and L_c_ increases first but then decreases.

The position and intensity of the absorption peak of the molecular vibration spectrum appearing in the infrared region depend on the vibration form of each group in the molecule and the influence of adjacent groups. Therefore, spectral analysis can be carried out as long as the vibration frequencies of various groups, i.e., the position of the absorption peak, and the displacement rule, are mastered. Huntjensf and Kreveen obtained structural information on the aromatic and aliphatic elements in the molecular structure of coal for the first time using XRD and FTIR techniques [20]. Subsequently, FTIR technology has often been used to study the functional group distribution, metamorphic degree, lithofacies composition, and genetic type of coal [21,22,23,24,25]. Davis et al. used FITR technology to analyze the infrared spectrum band of the stretching vibration of the aromatic structure in coal [26]. FTIR studies of some coals have shown that with the increase of coal metamorphism, the aromatization degree of coal increases, while the aliphatic and oxygen-containing functional groups decrease [20,23,27].

The ^13^C NMR spectrum can be used to directly obtain information on the carbon skeleton of coal. When coal is directly measured, errors caused by solvent extraction and incomplete extraction can be eliminated. However, ^13^C NMR has a low signal-to-noise ratio and sensitivity. Fourier transform (FT), cross polarization (CP), and magic angle rotation (MAS) must be used to improve its sensitivity. Sullivan and Maciel used dipole phase shift and cross polarization/magic angle rotation (CP/MAS) ^13^C NMR techniques to quantitatively analyze protonated carbon and non-protonated carbon [27]. Wang et al. [28] used ^13^C NMR techniques to analyze the types and existing forms of aromatic structures and aliphatic structures of lignite, combined with elemental analysis, and provided the molecular structure model of lignite. The structural parameters of coal molecules, including the composition and content of functional groups, the aromatic carbon ratio, and the aromatic ring condensation degree, can be quantitatively obtained by ^13^C NMR and FTIR techniques. A molecular structure model of coal is constructed by combining quantum chemical calculation methods [29].

The structure type of coal can be studied by electron microscopy, and the relationship between the material structure of coal and its formation process can be understood. Scanning electron microscopy (SEM) is used to determine the size, shape, quantity, distribution, and semi-quantitative composition and combination of mineral particles in coal [30,31,32]. Relevant research has shown that the brittleness of coal remains for micron-sized particles. Fissures are located between molecules and often traverse molecules [33]. Traditional research methods have great limitations for the study of coal’s supra-structure. Ordinary electron microscopes (optical microscopes and SEM) cannot reach the nano-scale observation scale, while the low-temperature liquid nitrogen adsorption method and mercury intrusion method cannot directly and comprehensively describe the nano-scale pore structure characteristics of coal, so it is necessary to directly observe the nano-scale microstructure morphology of coal and qualitatively and quantitatively analyze the microstructure parameters. Atomic force microscopy (AFM) has the characteristics of convenient operation and simple sample preparation. It can directly image the sample in a natural state to generate clear and repeatable high-resolution nano-images. More importantly, it can intuitively and comprehensively present three-dimensional dynamic images of the sample, which represents great progress in the nano-scale research of coal. Golubev et al. [34] described the supra-molecular evolution mode of natural asphaltenes under the observation of AFM, and proposed the importance of a supra-molecular structure in defining the division of solid asphaltenes. Lawrie et al. [35] measured the porosity of macerals (vitrinite and inertinite) of coal by AFM, and found that pores larger than 8 nm exist between the contact points of spherical molecular clusters and are also “channels” in the coal body caused by physical changes of coal.

By reviewing important achievements made in the study of coal’s molecular structure by physical methods, the use of different analysis techniques in the characterization of coal structure can be deeply understood. Analyzing the X-ray diffraction peak of coal’s crystal structure can produce important information of its crystal arrangement structure and aromatic layer structure. The composition information of macromolecular functional groups of coal can be obtained by analyzing the FTIR spectra of coal. ^13^C NMR technology provides connection information on basic structural units for the establishment of coal’s molecular skeleton structure.

The use of spectroscopy to characterize the molecular structure of coal has different values in different disciplines. In the field of the coal chemical industry, attention is paid to the composition and distribution of macromolecular functional groups of coal. In the field of coal geology, attention is paid to the relationship between the order degree of coal’s structure and the degree of coalification. In the field of coal processing and utilization, attention is paid to the relationship between the basic properties and the molecular structure of coal. Even the same technology has different concerns in different fields. In this study, we tried to depict different information on coal’s molecular structure through a number of analysis techniques, and conducted a superposition analysis of various experimental data to clarify the characteristics of Huainan coal’s molecular structure. In addition, spectral analysis is a very detailed methodology. The analysis process includes data denoising, smoothing, baseline removal, peak finding, fitting, etc. In order to obtain relatively accurate and reliable analysis data, each process needs to be rigorous to reduce man-made errors as much as possible.

Huainan coalfield has 44.4 billion tons of coal reserves, accounting for 19% of China’s coal reserves. At present, the annual output of coal is 40 million tons, which is reputed as “China’s energy capital”. Huainan coal has the characteristics of a low sulfur content, low phosphorus content, high volatility, high calorific value, rich oil, etc. It is an ideal raw material for the power coal and coal chemical industry, and is the industrial granary and power hometown of East China. Therefore, Huainan coal is the research object in this study, and a quantitative characterization of its macromolecular structure was comprehensively carried out by using XRD, FTIR, ^13^C-CP/MAS NMR, and other experimental methods (SEM/AFM). The research results are helpful for the extensive processing and efficient and clean utilization of Huainan coal. The technical process of this study is shown in Figure 1. This study is the first step in a series of studies and is also important basic research for answering questions regarding the macromolecular structure of Huainan coal. On the basis of this research, our subsequent research is based on the chemical vapor deposition (CVD) method, with the aim of carrying out the preparation of Huainan coal-based graphene and product performance testing, and the relevant results in this area are currently being collated.

## 2. Results and Discussion

### 2.1. SEM Characteristics of Original HNC

Previous classification of coal pores mainly includes the classification of micrometer-sized pores. Typical micro-scale pore classification schemes, such as the genetic classification scheme of Gay et al. [36], divide reservoir pores into intermolecular pores, coal-forming plant tissue pores, thermal genetic pores, and fracture pores. Thermogenic pores are traces left by gas generation and escape during coalification. They are mostly round in shape and have a pore size of approximately 1 μm. Plant tissue pores are between 10 and 1 μm. The pore size distribution and morphological characteristics of intermolecular pores and fracture pores are quite different. In the observation of the HNC sample in this paper, these micron-sized pores were also developed. The main types of developed micron-sized reservoir spaces include thermogenic pores with different development pore sizes (Figure 2a,b), fractures (Figure 2c,d), and plant tissue pores (Figure 2f). According to the cause of the formation of micro-pores in coal and the observation results of SEM, it is believed that HNC mainly develops the induced pores, which are thermogenic pores formed during coal coalification. Pore structures with a cross-sectional area of 10–300 μm^2^ and a certain number of pores with a diameter of 3–50 μm can mainly be observed under SEM. Micro-fractures are widely developed in coal. The dominant occurrence and microscopic components of micro-fractures are homogeneous vitrinite and structural vitrinite with scattered cell residual pores. Based on SEM observations, the length of micro-fractures varies, most of which are concentrated between 20 and 200 μm; the variation range of fractures width is obviously between 1 and 15 μm, and micro-fractures are mostly developed in parallel and perpendicular to the plane. On the whole, in HNC samples, the number of fractures is less developed, the width is smaller, the connection between fractures and fractures is weaker, and the clay minerals on the surface of coal are mostly irregular lamellate montmorillonite. In addition, a clear lamellar structure can be observed in Figure 2e. Figure 2f shows the cross section of organic matter. These micron-sized pores, fractures, and structures form the microscopic storage space of coal reservoirs to a great extent.

### 2.2. AFM Characteristics of Original HNC

SEM observation of nano-scale pore and fracture structures on coal’s surface is not ideal, but AFM technology can achieve this objective. Figure 3a presents an HNC high-resolution two-dimensional AFM image with an image range of 1000 nm × 1000 nm. The difference in chromatographic color in this figure represents the composition of different substances. Nanowires (green protrusions in the 2D plan) formed by coal molecular group stacking and aggregation show the gridded structure as a whole. The bright and dark alternate phenomena with nano-wires are the pores and fractures (crimson depressions in the 2D plan) between the main chain skeletons of coal molecules and the small molecular matrix embedded in coal (dark blue lines in the 2D plan). The intermolecular pore morphology of the HNC sample is mainly round and elliptical. In general, the length of the short axis is about 80 nm and the length of the long axis is about 140 nm. The surface fractures are less developed and the length is mostly about 750 nm. The macromolecular structure of coal determines the development degree of intermolecular micropores in coal [35]. HNC is a gas-fat, low-grade metamorphic coal, so its has a low degree of molecular aromatization, relatively developed side chains and functional groups, a large molecular radius, relatively loose accumulation of macromolecules, not enough tight combinations between structural units, and a certain number of pores between molecules. These pores are mainly caused by the thermal evolution of coal and can be attributed to interchain pores in terms of the chemical structure [30]. Using the 3D Image function in Nanoscope Analysis software, 3D rendering of 2D images can be realized, so the spatial distribution of coal’s molecular structure can be visually observed. Figure 3b shows the gridded arrangement of HNC’s macromolecular structure, and the green bands are the basic structural units of coal’s macromolecular clusters. The length of molecular groups formed by macromolecular accumulation is mostly between 200 and 350 nm. The dark regions adjacent to molecular groups are pores and fractures.

### 2.3. XRD Characteristics of HNC before and after Acid Treatment

XRD patterns of raw coal and demineralized coal are shown in Figure 4. In the XRD pattern of raw coal, the diffraction peaks of kaolinite and quartz are obvious, and chlorite mineral diffraction peaks are found. However, there is a microcrystalline carbon (002) diffraction band between 24° and 26°, which is completely covered by minerals in the original XRD pattern. In addition, by comparing the diffraction patterns of HNC samples before and after demineralization, it can be seen that the diffraction background value of raw coal is relatively high and the baseline is jagged, which may be related to the low cleanliness of the main mineral kaolinite.

The demineralized HNC was subjected to peak fitting treatment, and see Section 3.9.1 for the calculation of structural parameters *d_002_*, *L_a_*, *L_c_*, and *N_ave_* of aromatic lamellae in HNC.

Previous studies have confirmed that there are nine obvious diffraction peaks in the diffraction spectrum of graphite, indicating that it is a crystal arrangement structure. The diffraction peak of the X-ray diffraction spectrum of coal is not as fine as graphite, and the diffraction intensity is not as good. However, some diffraction peaks are still found, indicating that some ordered carbon does exist in coal. HNC has the two main diffraction peaks (002) and (001), corresponding to graphite. Figure 4 shows two obvious spectral bands: one is a peak band with 2θ between 10° and 35°, i.e., the (002) diffraction peak band, and the other is a peak band with 2θ between 35° and 50°, i.e., the (100) diffraction peak band. The structural parameters of the (002) and (100) diffraction peaks could be obtained by fitting the two peak bands separately (Figure 5). It can be seen from Figure 5a that there is a sub-peak on the left side of the (002) spectral peak, i.e., the λ peak. The size of this peak area reflects the degree of stacking (degree of ordering) of aromatic lamellae in coal, i.e., the larger the area, the lower the degree of ordering. It can be seen that the orientation degree of HNC aromatic ring carbon network lamellae in the space arrangement is relatively low. Based on the peak fitting results and combined with several Equations described in Section 3.9.1, d*_002_*, L*_c_*, and N*_ave_* could be calculated (Table 1).

Figure 5b shows the results of peak fitting in the (*100*) band. It can be seen that there are four sub-peaks in 2*θ* around 44°–45°, and these peaks have a certain diffraction intensity, which further reflects that HNC’s aromatic carbon network structure is indeed highly disordered. In this study, in order to comprehensively reflect the characteristics of the (*100*) band, the structural parameters of the four sub-peaks were calculated by superposition (Table 1).

As can be seen from Table 1, the L_a_/L_c_ value of HNC is close to 1:1, and N_ave_ reveals that the effective stacking number of aromatic layers is about 7. Combined with the coal structure model proposed by Gay et al. [36], we present a schematic diagram of HNC’s structure (embedded in Figure 4b). It can be seen that the coal aromatic layer structure of HNC is similar to a cube structure in three-dimensional space, but due to the low condensation degree of aromatic lamellae, the long branched chains around the carbon skeleton affect the orientation degree of the aromatic layer.

### 2.4. Infrared Spectral Characteristics of HNC and Its Oxygen-Containing Functional Groups

#### 2.4.1. FTIR Characteristics of HNC after Acid Treatment

Figure 6 presents an infrared spectrum of HNC. As can be seen from the figure, 1000–300 cm^−1^ displays a sharp peak; from 2000 to 1000 cm^−1^, there are two sharp peaks and one convex peak. 4000–2500 cm^−1^ exhibits a sharp peak and an obviously convex peak. Ibara et al. gave the detailed analysis process of the infrared spectrum of coal and the assignment of the absorption peak of the infrared spectrum [37]. It should be pointed out that not all spectral peaks can be associated with chemical structures, especially fingerprint regions. In many cases, the analysis of the infrared spectrum needs to be based on experience, because the vibration frequency of chemical bonds is quite sensitive to the surrounding chemical environment. Therefore, the research results of Ibara et al. are comprehensively considered in this study [37], and the infrared spectrum characteristics of HNC are analyzed in combination with the actual observation results.

The detailed analysis of Figure 7 shows that there are multiple absorption peaks at 400–500 cm^−1^, which represent substituent groups of aromatic rings. There are sharp absorption peaks at 650 and 750 cm^−1^, which show the stretching vibration of mono-substituted or 1, 3-substituted aromatic hydrocarbon –CH. There are several absorption peaks at 1000–1300 cm^−1^, which may be attributed to the absorption of ether or functional groups without carboxyl and methoxy groups. The symmetrical bending vibration absorption peak with –CH_3_– of 1435 cm^−1^ indicates that saturated aliphatic side chains such as –CH_3_ or –CH_2_– are expected to exist in the structure. There is a strong absorption peak at 1600 cm^−1^, which is mainly the result of the overlapping absorption of the C=C double bond of the aromatic ring and hydrogen-bonded carbonyl. There is a weak C=O absorption peak at 1720 cm^−1^, which indicates that the structure may contain carbon-oxygen unsaturated double bonds, such as carbonyl or carboxyl. The strong and sharp absorption peak at 2918 cm^−1^ may represent the antisymmetric stretching vibration of the saturated carbon atom –CH_2_–. There is an unsaturated C–H stretching vibration absorption peak at 3050 cm^−1^. There is a broad absorption peak at 3390 cm^−1^, which is mainly represented by the stretching vibration of O–H or N–H associated with intermolecular hydrogen bonds.

The infrared spectra of 1800–1000 cm^−1^ were fitted (Figure 7a), and the assignment of each absorption peak was determined (Table 2). It can be seen that the forms of oxygen in the molecular structure of HNC are mainly ether, phenol, and carbonyl. The ratio of ether (R–O–R’), phenol (ArOH) and carbonyl groups (C=O) is about 6:1:2, as obtained through a superposition calculation of the area of each peak position. Similarly, fitting of the infrared spectra of 3000–2800 cm^−1^ (Figure 7b) shows that the aliphatic chains in HNC’s molecular structure are mostly methyne and methylene, and the ratio of methyl (–CH_3_), methyne (≡CH), and methylene (–CH_2_–) is about 1:5:3, as obtained through a calculation of the peak area (Table 2).

#### 2.4.2. Calculation of Oxygen-Containing Functional Group Content

Oxygen in coal can exist in the form of moisture, inorganic oxygen-containing compounds, and oxygen-containing functional groups (OCFGs), of which oxygen in the form of OCFGs has a greater impact on the properties of coal. The OCFGs in coal are divided into carbonyl, carboxyl, hydroxyl (phenolic hydroxyl), ether oxygen, methoxy, and other types. Van Krevelen demarcates OCFGs in coal into hydroxyl (phenolic hydroxyl), carboxyl, methoxy, carbonyl and inactive oxygen [38]. The OCFGs in coal can be obtained by studying the solvent extraction of coal or depolymerization products through pyrolysis, hydrolysis, oxidation, and other processes. The research methods mainly include empirical formula calculations, selective reagent reactions, and instrumental analysis. In this study, the OCFG content of HNC was calculated by an empirical formula.

On the basis of the ultimate analysis results of HNC (Table 3), the content of the S_daf_ element is 0.35%, so the empirical formula generalized by Attar can be used to calculate the content of OCFGs in coal [5]. Attar expressed the relationship between the nature and structure of coal using the Moore additive function [5]:(1)$$MW\;=\sum X_{i}\cdot MW_{i}=\;C\cdot X_{c}+H\cdot X_{H}+O\cdot X_{O}+N\cdot X_{N}\;\left(i=\;C,\;H,\;O,\;N\right),$$
where $X_{c}$, $X_{H}$, $X_{O}$, and $X_{N}$ are the mole fractions of C, H, O, and N, respectively, and *MW* is the molecular weight.

From Equation (1), it can be deduced that
(2)$$X_{i}=\frac{N_{i}/MW_{i}}{C_{daf}/12+H_{daf}/1+N_{daf}/14+O_{daf}/16},$$
where $N_{i}$ is the content of C_daf_, H_daf_, O_daf_, and N_daf_ in coal, and $MW_{i}$ is the atomic weight of C, H, O, and N.

Therefore, the number of moles of the j OCFGs per 100 moles of carbon can be defined as $O_{j}$:(3)Oj=∑Oj=100×XOXC,
where j=−O−,−C=O,R−OH ,−COOH.

It can be ascertained that the oxygen content $Z_{j}$ in a certain functional group can establish the following relationship with the variable u:(4)$$u={\left(100X_{O}/X_{C}\right)}^{2}\cdot \left(1/MW\right),$$
(5)$$Z_{j}=100O_{j}\cdot {\left(X_{C}/100X_{O}\right)}^{2}.$$

Then, the mathematical formulas for calculating OCFGs can be expressed as
(6)$$\mathrm{Log}Z_{-O-}=1.2512-0.6483Logu,$$
(7)$$\mathrm{Log}Z_{-C=O}=0.4493-0.4334Logu,$$
(8)$$\mathrm{Log}Z_{-OH}=0.3186-0.6043Logu,$$
(9)$$Z_{-COOH}=0.1835+0.0185u-0.000119u^{2}.$$

According to Equations (6)–(9), the OCFG content of HNC calculated is listed in Table 3. It can be seen from Table 3 that the oxygen in HNC is mainly ether oxygen (–O–), followed by carbonyl (–C=O), and contains a small amount of phenolic hydroxyl (–OH), and the ratio of the three is approximately 8:1:2, which is similar to that obtained by FTIR.

In addition, when the sulfur content is negligible, the estimated molecular structure C_100_H_76_O_9_N of organic matter in HNC can be obtained from the molar fraction of elements in Table 3.

#### 2.4.3. NMR Characteristics of HNC

The ^13^C-CP/MAS NMR spectrum of HNC is shown in Figure 8. According to the research results of Trewhella et al. [38], the structural assignment of chemical shifts in the spectrogram is listed in Table 4. The spectral peak parameters obtained by PeakFit software in different chemical shift intervals are also listed in Table 4. As can be seen from Figure 8, the ^13^C NMR spectrum of the sample can obviously divided into three peak regions: one is the aliphatic carbon region, with a chemical shift of 0–90; the second is the aromatic carbon region, with a chemical shift of 90–165; and the third is the carbonyl peak region, with a chemical shift of 165–220. It can also be seen that the intensity of the peak in the aromatic region is much larger than that in the aliphatic region, indicating that aromatic carbon atoms are the main components in the macromolecular structure of HNC, while aliphatic carbon atoms play a role in linking these aromatic structural units in the structure. Combined with the XRD analysis results of HNC (Table 1), it can be concluded that the carbon network structure of aromatic lamellae is formed by stacking aromatic carbon atoms, while the branched carbon-containing groups are composed of aliphatic carbon atoms.

In order to study the relative content of carbon atoms with different structures, Solum et al. proposed 12 structural parameters of coal on the basis of previous research work [39], including $f_{a}$ (aromatic carbon ratio), $f_{a}^{C}$ (carbonyl carbon content with chemical shift > l65), $f_{a}^{\prime }$ (ratio of *sp*^2^ hybrid carbon in the aromatic ring to total carbon), $f_{a}^{H}$ (protonated carbon), $f_{a}^{N}$ (non-protonated carbon), $f_{a}^{P}$ (phenol or arylether carbon, 150–160), $f_{a}^{S}$ (substituted aromatic carbon, 135–150), $f_{a}^{B}$ (aromatic bridgehead), $f_{al}$ (aliphatic carbon ratio), $f_{al}^{H}$ (quaternary carbon, CH, and CH_2_ groups), $f_{al}^{*}$ (lipomethyl and arylmethyl), and $f_{al}^{O}$ (bonded to oxygen).

In the CP/MS experiment, the aromatic carbon rate $f_{a}$ and the aliphatic carbon rate $f_{al}$ could be calculated according to the initial values of the magnetization vectors of the aromatic carbon and the aliphatic carbon, i.e., $f_{a}=\frac{M_{\left(0\right)ar}}{M_{\left(0\right)ar}+M_{\left(0\right)al}}$ and $f_{al}=\frac{M_{\left(0\right)al}}{M_{\left(0\right)ar}+M_{\left(0\right)al}}$.

$f_{a}$ can be composed of $f_{a}^{\prime }$ and $f_{a}^{C}$, and can be expressed as follows: $f_{a}=f_{a}^{\prime }+f_{a}^{C}$.

The Dipole dephasing technique can be used to distinguish carbon atoms with different effective dipolar action intensities, and $f_{a}^{\prime }$ can be partitioned into protonated carbon $f_{a}^{H}$ and non-protonated carbon $f_{a}^{N}$, i.e., $\;f_{a}^{\prime }=f_{a}^{N}+f_{a}^{H}$. The simplified calculation method proposed by Wilson et al. was used in this study to calculate the relative content of protonated carbon [40]:(10)$$f_{a}=\frac{1.22\times 0.88\times M+P}{T}$$
where M is the aromatic carbon signal intensity when the delay time *t* = 40 *μ*s, P is the aromatic carbon signal intensity connected to protons when the delay time *t* = 0 *μ*s, and T is the total carbon signal intensity when the delay time *t* = 0 *μ*s.

According to Rayleigh fractionation, van Krevelen believes that the volatile matter of coal is converted from substances other than aromatic hydrocarbons [41], and has put forward a calculation formula for the volatile matter of coal. Based on this, van Krevelen has given the calculation formula for the aromatic carbon rate of coal ($f_{a}$):(11)$$f_{a}=\frac{\left(100-V_{daf}\right)\times 1200}{1240C_{daf}}$$

$f_{a}$ calculated according to Equation (11) is substituted into Equation (10), and M and T can be obtained through experiments (M8T24=0.33), thus producing P. The relative content $f_{a}^{H}$ of protonated aromatic carbon can be obtained by the following formula: $f_{a}^{H}=P⁄T$. Furthermore, the relative content $f_{a}^{N}$ of non-protonated carbon can be obtained.

Non-protonated carbon can be divided into three types ($f_{a}^{P}$, $f_{a}^{S}$, and $f_{a}^{B}$), according to the chemical shift interval. Among them, $f_{a}^{P}$ and $f_{a}^{S}$ can be calculated by NMR spectrum peak area integration (see Table 4), while $f_{a}^{B}$ can be calculated by the following formula:(12)$$f_{a}^{B}=f_{a}^{N}-f_{a}^{P}-f_{a}^{S}$$

In the ^13^C NMR spectrum, the aliphatic carbon region can be subdivided into $f_{al}^{H}$, $f_{al}^{*}$, and $f_{al}^{O}$, that is, $f_{al}\approx f_{al}^{H}+f_{al}^{*}+f_{al}^{O}$. Their relative contents could be obtained by accumulating the area ratio of fitting peaks in each interval (see Table 4). Then, the aliphatic carbon ratio $f_{al}$ was calculated.

Based on the above parameter calculation, some important parameters of coal structures could be deduced. For example, the aromatic hydrogen ratio ($H_{a}$) could be calculated from $H_{a}={\left(X_{c}/X_{H}\right)}_{mole}\times f_{a}^{H}$, which reflects the degree of hydrogen enrichment in aromatics. The average structural size of aromatic nuclei ($X_{b}$) could be calculated from $X_{b}=f_{a}^{B}/\left(f_{a}^{H}+f_{a}^{P}+f_{a}^{S}\right)$, which indicates the size of aromatic clusters. According to the above detailed calculation procedure, the structural parameters of HNC are listed in Table 5.

Vandenbroucke and Largeau demonstrated that, in the degradation stage of coal (deep metamorphism) (vitrinite maximum reflectance is 0.5–2%) [42], the condensation degree of aromatic ring is 2–4 rings. The average structural size of HNC’s aromatic nucleus is 0.16 ($X_{b}$), while the ratio of aromatic compound naphthalene is 0.25, which is 1.5 times that of HNC. That is to say, naphthalene, an aromatic compound with two aromatic rings, can already represent the aromatic structural characteristics of HNC, while the rest are aromatic rings composed of benzene rings and heteroatoms. In addition, information on arylmethyl (Ar–CH_3_), methyne (≡CH), and methylene (–CH_2_–) in HNC’s molecular structure can be seen in Table 4. The area ratio of the Ar–CH_3_ assignment peak is ~0.033 (chemical shift, 22.8), ≡CH is ~0.050 (chemical shift, 37.1), and –CH_2_– is ~0.047 (chemical shift, 29.6). The ratio between the three is approximately 2:3:3, indicating that HNC is relatively rich in ≡CH and –CH_2_– structures. This is somewhat different from the FTIR results (1:5:3), mainly concentrated on Ar–CH_3_ and ≡CH. The content of Ar–CH_3_ obtained by FTIR is lower than that obtained by NMR, while that of ≡CH is higher. This may be related to the accuracy of different experimental tests and analyses, and is worth further study.

## 3. Materials and Methods

### 3.1. Sample Collection

In this study, the coal samples analyzed were from the No. 13-1 coal seam of the Upper Shihezi Formation in Huainan coalfield, northern China (denoted as HNC). In order to ensure the freshness of samples, groove sampling was carried out in the underground working face. In the laboratory, vitrified coal strips (removing gangue) were obtained by a manual separation method, and they were ground and sieved until the particle size reached 200 mesh. The ground powder sample was then dried and stored in a sealed bag for testing.

### 3.2. Proximate Analysis and Ultimate Analysis

The ultimate analysis of HNC samples was carried out on a Vario EL element analyzer of German EA Company. The contents of carbon (C_daf_, %), hydrogen (H_daf_, %), nitrogen (N_daf_, %), and sulfur (S_daf_, %) were the average of two parallel samples, and the oxygen (O_daf_, %) content was obtained by difference subtraction. According to the GB/T212-2008 standard, the moisture (M_ad_, %), ash (A_ad_, %) and volatile matter (V_daf_, %) of HNC samples were determined.

### 3.3. Demineralization of Samples

In recent years, some researchers have thoroughly investigated the influence of inorganic mineral compositions in coal on the spectral measurement results of coal’s chemical structure, and believe that inorganic mineral compositions have a certain degree of interference in the fine observation of coal’s chemical structure. In order to avoid the influence of inorganic mineral compositions on the XRD, FTIR, and ^13^C NMR experimental results, the whole rock powder samples should be demineralized. The specific demineralization process was documented in our previous research results [36]. The mineral composition of coal before acid treatment can be characterized by XRD and FTIR. However, considering the experimental conditions and the degree of sample preservation, XRD was used in this study to analyze the minerals of coal before demineralization.

### 3.4. XRD

A Philips X’Pert PRO X-ray diffractometer (XRD) was used to record X-ray intensities scattered from the HNC samples. Cu Kα radiation (60 kV, 55 mA) was used as the X-ray source. Powdered coal samples were packed into a rectangular cavity in an aluminum holder and scanned from 10° to 70° in the 2θ range, with a 0.1° step interval and a 2 s/step counter time.

### 3.5. FTIR

Quantitative FTIR transmission spectra analysis was conducted on coal KBr pellets. The pellets were prepared by mixing 1 mg of demineralized HNC with 200 mg of KBr for 5 min and then pressing the mixture into pellets in an evacuated die under 10 MPa of pressure for 2 min. The pellets were dried in a vacuum oven for 24 h to exclude the interference of water in the spectrum. The infrared spectra were generated by collecting 128 scans at a resolution of 8 cm^−1^ using a Nicolet model 8700 Fourier Transform Infrared spectrometer. The measured region extended from 400 to 4000 cm^−1^.

### 3.6. ^13^C-CP/MAS NMR

The ^13^C NMR experiment of HNC was carried out on a Varian INOVA300 superconducting NMR instrument. A solid double resonance probe was used, with an outer diameter of 6 mm for the ZrO_2_ rotor, magic angle rotation speed of 6 kHz, ^13^C detection nucleus resonance frequency of 75.425 MHz, sampling time of 0.05 s, pulse width of 4 μs, cycle delay time of 4 s, and number of scans of 7000. Cross polarization (CP) technology was used. The contact time was 5 ms and the spectrum width was 3000 Hz.

### 3.7. SEM

The JSM-6490LV scanning electron microscope was used to observe the surface morphology of coal samples. Before observation, Ar ion polishing technology was used to treat the surface of HNC, and the sample presented a microscopic observation area of about 2 mm^2^. The significant performance parameters included tungsten filament lighting, a point resolution of 3 nm, an acceleration voltage of 0.5~30 kV, and magnification of ×5~300,000.

### 3.8. AFM

A NanoScope AFM manufactured by Digital Company (SPA-300HV, Chiba, Japan) was used to provide insight into the morphological characteristics of HNC. The maximum scanning range was 30 μm × 30 μm × 2 μm. The automatic scanning surface groove was 1800/mm, which can offer a better resolution of the topography of the coal surface. The information on sample preparation and experimental conditions can be found in our previous research results [43]. The scanning images were interpreted by Nanoscope Analysis software.

### 3.9. Data Processing and Calculation Procedure

#### 3.9.1. XRD

XRD patterns of HNC samples were fitted by Origin 2017 software. GaussAmp was selected as the fitting function model. According to the peak position (diffraction angle, *θ*) and half-peak width (*β*) parameters of the fitting peak, the molecular structure parameters of HNC were calculated respectively.
(13)$$d_{002}=\frac{\lambda }{2\sin\theta_{002}},$$
(14)$$L_{c}=\frac{K_{1}\lambda }{\beta_{002}\cos\theta_{002}},$$
(15)$$N_{ave}=\frac{L_{c}}{d_{002}},$$
(16)$$L_{a}=\frac{K_{2}\lambda }{\beta_{100}\cos\theta_{100}},$$
where *d_002_*, *L_a_*, *L_c_*, and *N_ave_* were used to calculate the distance between aromatic ring plies, the diameter of aromatic ring plies, the stacking height of aromatic ring plies, and effective stacking layers of aromatic ring plies, respectively. *λ* represents the X-ray wavelength (nm). This experiment used copper targets, and *λ* = 1.54056 Å. *θ*_002_ and *θ*_00l_ represent diffraction angles (°) corresponding to the peak positions of (002) and (100), respectively. *β_002_* and *β_100_* represent the peak width at half-height values (rad) of (002) and (100) peaks, respectively. K_l_ and K_2_ are microcrystalline shape factors, where K_1_ = 0.94 and K_2_ = 1.84.

#### 3.9.2. FTIR

The infrared spectra of HNC samples were fitted by Origin 2017 software. GaussAmp was selected as the fitting function model. The model provided the parameters of the peak position (*x_c_*), peak area (*A*), and full width half maximum (FWHM), etc. The relative contents of different chemical groups could be obtained by calculating the parameter (*A*) of the fitted peaks.

#### 3.9.3. ^13^C NMR

The ^13^C NMR spectra of HNC samples were fitted by PeakFit software (developed by AISN Software), and GaussAmp was selected as the fitting function model. The fitting spectrum can be divided into the aliphatic carbon region, aromatic carbon region, and carbonyl and carboxyl peak region. The structural parameters were obtained by using the peak fitting results.

#### 3.9.4. SEM

In order to obtain the size of pores and fractures in the image, SmileView software was used for SEM image analysis.

#### 3.9.5. AFM

Firstly, the 2D Image Function in Nanoscope Analysis software was employed to realize the 2D analysis of images. When running this function, the original AFM image of HNC was first filtered to improve the pixel quality of the image. By adjusting the chromatographic column, a clear image of coal’s macromolecular clusters and their surface pore structures could be obtained. Secondly, using the 3D Image Function, the surface morphology of coal was presented from a three-dimensional perspective, and the composition of coal’s macromolecular structure was further observed. Please find the original data of XRD, FTIR and NMR test results of HNC in Appendix A.

## 4. Conclusions

Fine characterization of coal’s molecular structure model is of great theoretical value for further understanding coal’s combustion characteristics and improving the combustion efficiency. Therefore, people use different methods to study the structure of coal from different angles. However, due to the complexity of coal and the limitation of research methods, research on the macromolecular structure of coal still lacks systematicness. With the continuous development of the Huainan coal deep processing industry, it will be necessary to understand the fine structure of Huainan coal. In this study, a number of experimental methods were used to describe the structure of HNC. Some important understandings and conclusions have been obtained.

The SEM experimental results of HNC show that the high-resolution morphology of coal’s surface contains pores and fractures of different sizes and develops stepped bedding. AFM gives the loose arrangement mode of HNC’s macromolecular structure. Through fitting the XRD pattern of HNC, it is shown that the condensation degree of HNC’s aromatic ring is low ((100) spectral peak is obvious) and the orientation degree of carbon network lamellae is poor ((002) spectral peak is mixed with a sub-peak). The calculated ratio of the diameter of aromatic ring lamellae to their stacking height (L_a_/L_c_ = 1.05) and the effective stacking number of aromatic nuclei (N_ave_ = 7.3) show that the molecular space structure of HNC is a cube formed of seven stacked aromatic ring lamellae. By fitting the infrared spectral peaks of HNC, the attribution of diffraction peaks was obtained. The FWHM and peak area parameters of the fitted peaks were analyzed, and it was found that the fatty chains in HNC’s molecular structure were mainly methyl and methylene. Oxygen is mainly –O–, followed by –C=O, and contains a small amount of –OH, with a ratio of about 8:1:2. The approximate molecular structure C_100_H_76_O_9_N of organic compounds in HNC was obtained by the mole fraction of binding elements. The ^13^C NMR full-spectrum fitting of HNC was carried out, and the structural parameters of fitting peaks in different chemical shift attribution regions were obtained. It is recognized that in HNC’s structure (the average structural size of aromatic nucleus $X_{b}$ = 0.16), the form of aromatic carbon atoms is mainly naphthalene, with the condensation degree of 2, and the rest are aromatic rings composed of benzene rings and heteroatoms. In addition, HNC also has relatively abundant ≡CH and –CH_2_– structures.

## Figures and Tables

**Figure 1 molecules-25-02661-f001:**
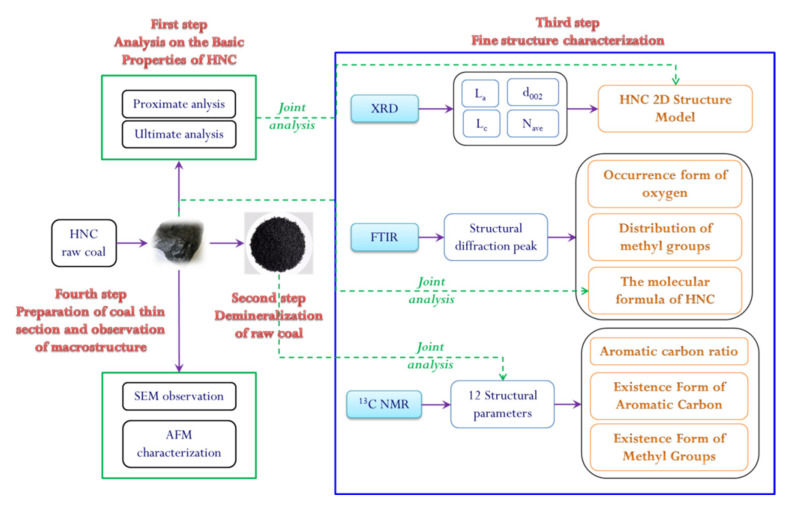
The analysis procedure chart of this study.

**Figure 2 molecules-25-02661-f002:**
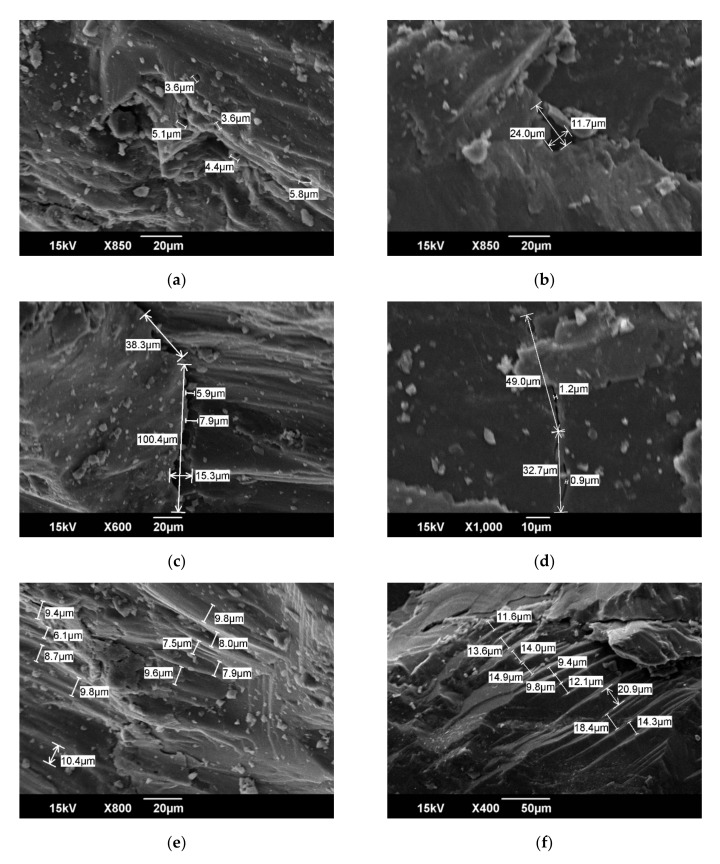
Micron-sized reservoir structure space developed in HNC. In (**a**), the average pore diameter of the thermogenic pores is 4.5 μm, while in (**b**), a thermogenic pore of 24 μm × 12 μm can be observed. In (**c**,**d**), there are some differences in the observed width development of endogenous fractures, and the former may be due to tensile stress. At ×800 (**e**), the average spacing of organic matter lamellae is ~8.7 μm; at ×400 (**f**), the average spacing between sections of organic matter is ~13.4 μm.

**Figure 3 molecules-25-02661-f003:**
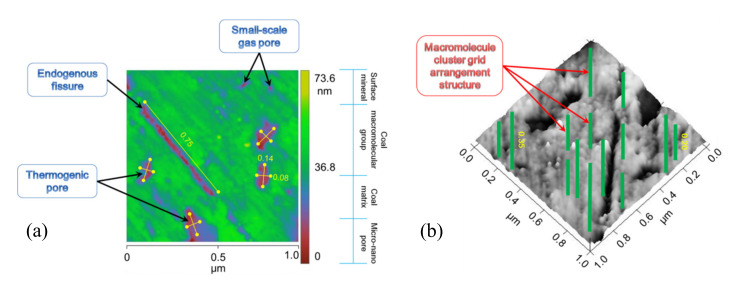
(**a**) 2D gridded arrangement structure formed by coal macromolecular groups. Green regions are nanowires formed by coal macromolecular accumulation, and dark regions between adjacent nanowires are intermolecular pores and fractures; (**b**) is a three-dimensional image of (**a**).

**Figure 4 molecules-25-02661-f004:**
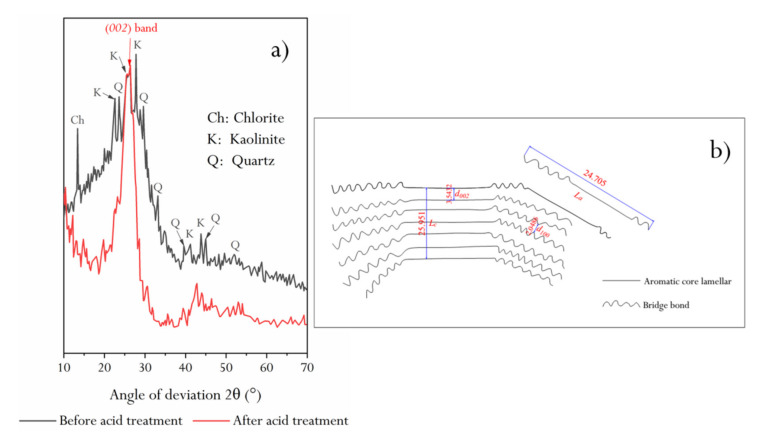
(**a**) XRD spectrum of HNC before and after acid treatment; (**b**) structural mode of HNC.

**Figure 5 molecules-25-02661-f005:**
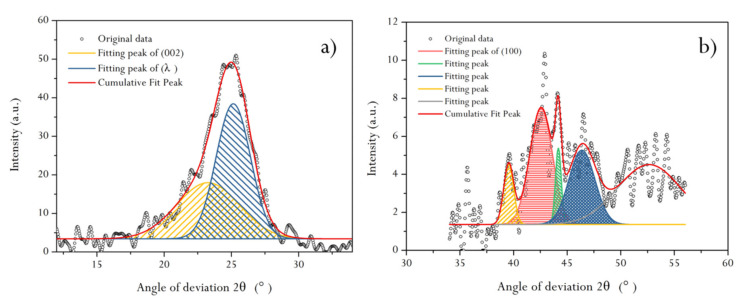
Peak fitting results of the demineralized HNC in (*002*) (**a**) and (*100*) (**b**) bands.

**Figure 6 molecules-25-02661-f006:**
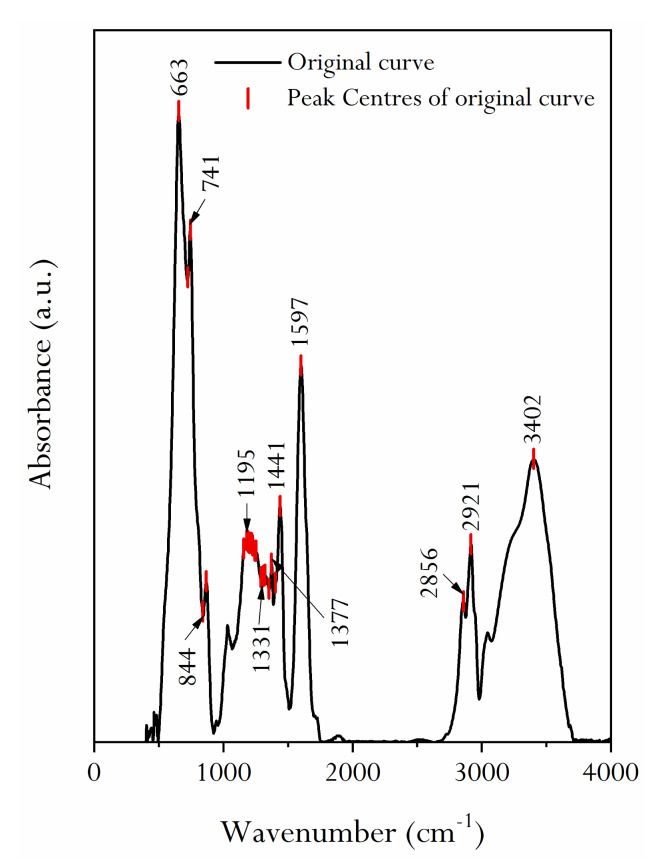
The infrared spectrum of the demineralized HNC.

**Figure 7 molecules-25-02661-f007:**
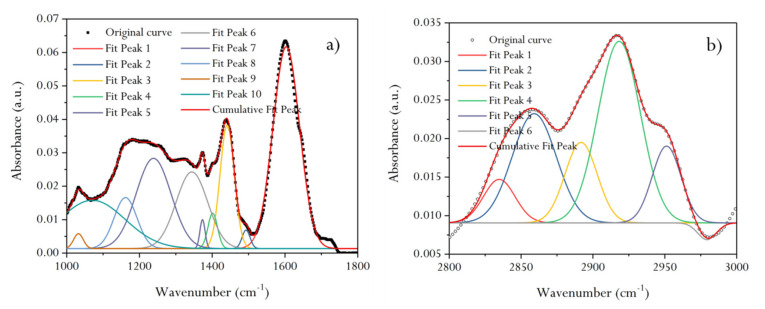
Infrared spectrum fitting diagram of the demineralized HNC in the 1000–1800 and 2800–3000 cm^−1^ band. (**a**) 10 sub-peaks are obtained by peak fitting in the 1800–1000 cm^−1^. (**b**) The fitting number of sub-peaks between 3000 and 2800 cm^−1^ is 6. The mathematical function model of fitting peak is Gaussian function.

**Figure 8 molecules-25-02661-f008:**
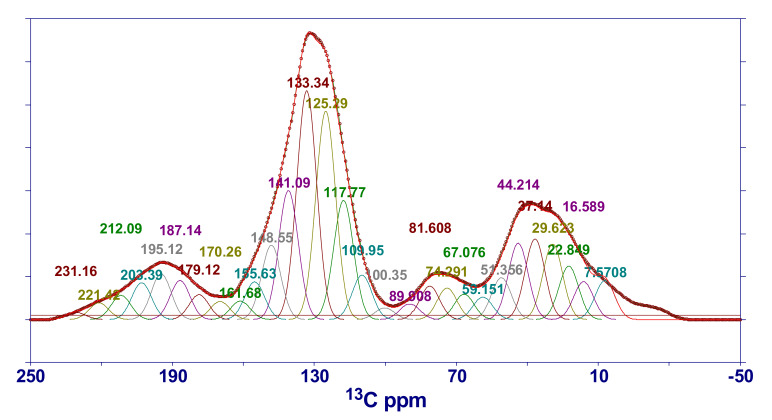
^13^C-CP/MAS NMR spectrum of the demineralized HNC. In the figure, the red solid line is the fitting curve, the black circle is the original data point, and the dotted line is the baseline.

**Table 1 molecules-25-02661-t001:** Calculation results of structural parameters of the demineralized HNC in (*002*) and (*100*) bands.

Band Interval	2*θ*_002_ (°)	*θ*_002_ (°)	Cos*θ*_002_	K_1_	*λ* (Å)	*β*_002_ (°)	*β*_002_ (rad)	L_c_ (Å)	d_002_ (Å)	d_100_ (Å)	L_a_/L_c_	N_ave_
10°–35°	25.11	12.556	0.976	0.94	1.54056	3.121	0.0545	25.951	3.5432	2.0486	1.0504	7.3242
Band interval	2*θ*_100_ (°)	*θ*_100_ (°)	cos*θ*_100_	K_2_	*λ* (Å)	*β*_100_ (°)	*β*_100_ (rad)	L_a_ (Å)				
35°–55°	44.17	22.086	0.926	1.84	1.54056	7.095	0.1238	24.705				

**Table 2 molecules-25-02661-t002:** Absorption peak parameters and assignment of the demineralized HNC in the 1000–1800 and 2800–3000 cm^−1^ band.

Spectral Interval (cm^−1^)	Fitting Peak Number	Peak Position (cm^−1^)	FWHM (cm^−1^)	Fitting Peak Area	Assignment	Absorption Intensity		
1000–1800	**1**	1603	85.95	5.522	aromatic nucleus (C=C)	S		
	2	1493	26.10	0.152	(C–C)_ar_ stretching			W
	3	1441	46.28	1.817	aliphatic chains (CH_3_, CH_2_)		M	
	4	1401	30.85	0.347	CH_3_−			W
	5	1373	16.66	0.153	CH_3_−Ar, R			W
	6	1344	102.54	2.501	CH_3_−C=O			W
	7	1238	113.49	3.260	C−O−C		M	
	8	1161	73.89	1.200	Hydroxybenzene		M	
	9	1032	31.05	0.147	Alkyl ether			W
	10	1070	212.45	3.267	C−O−C			W
2800–3000	1	2834	25.89	0.155	R_2_CH_2_		M	
	2	2859	36.24	0.546	R_2_CH_2_		M	
	3	2891	26.43	0.293	R_3_CH	S		
	4	2918	34.68	0.869	R_3_CH	S		
	5	2951	23.61	0.250	RCH_3_		M	
	6	2979	15.41	0.035	RCH			W

Note: S, strong; M, medium; W, weak.

**Table 3 molecules-25-02661-t003:** Basic properties of the demineralized HNC and content of oxygen-containing functional groups (OCFGs).

Analysis Object		Content
Proximate analysis (%)	M_ad_	1.37
	A_ad_	18.26
	V_daf_	33.82
Ultimate analysis (%)	C_daf_	83.3
	H_daf_	5.27
	O_daf_	9.85
	N_daf_	1.23
	S_daf_	0.35
OCFGs	X_C_	0.53748
	X_H_	0.40804
	X_O_	0.04766
	X_N_	0.00680
	Z_–O–_	3.957
	Z_–C=O_	1.028
	Z_–OH_	0.512
	Z_–COOH_	0.189

**Table 4 molecules-25-02661-t004:** Assignment of the chemical shift interval and fitting peak parameters (including the calculation of the main structural parameters) in the ^13^C-CP/MAS NMR spectrum.

Chemical Shift Interval	Assignment	Peak Position of Fitting Peak		Peak Area of Fitting Peak (%)	Peak Position of Fitting Peak		Peak Area of Fitting Peak (%)
16	R–CH_3_	$f_{al}^{*}$	7.6	0.024	117.8		0.074
20	Ar–CH_3_		16.6	0.024	125.3		0.129
23	CH_2_–CH_3_		22.8	0.033	133.3		0.141
33	CH_2_	$f_{al}^{H}$	29.6	0.047	$f_{a}^{S}$	141.1	0.080
36–50	≡CH		37.1	0.050		148.5	0.046
50–60	O–CH_3_, O–CH_2_		44.2	0.047	$f_{a}^{P}$	155.6	0.023
60–70	O–CH	$f_{al}^{O}$	51.4	0.025	161.7		0.011
75–90	R–O–R		59.2	0.014	$f_{a}^{C}$	170.3	0.011
100–129	Ar–H	67.1		0.016		179.1	0.015
129–137	Bridgehead (C–C)	74.3		0.019		187.1	0.024
137–148	Ar–C	81.6		0.021	195.1		0.028
148–165	Ar–O	89.9		0.010	203.4		0.023
165–190	COOH	100.4		0.007	212.1		0.015
190–220	C=O	110.0		0.028	221.4		0.010
					231.2		0.004

**Table 5 molecules-25-02661-t005:** The ^13^C-CP/MAS NMR structural parameters of HNC.

Structural Parameters	$f_{a}$	$f_{a}^{C}$	$f_{a}^{\prime }$	$f_{a}^{N}$	$f_{a}^{H}$	$f_{a}^{P}$	$f_{a}^{S}$	$f_{a}^{B}$	$f_{al}$	$f_{al}^{\prime }\;$	$\Delta f_{al}$	$f_{al}^{*}$	$f_{al}^{H}$	$f_{al}^{O}$	$X_{b}$	$H_{a}$
Calculated value	0.768	0.050	0.718	0.248	0.470	0.023	0.126	0.099	0.2315	0.2312	0.0003	0.049	0.143	0.038	0.160	0.620

Note: The sum of the aromatic carbon ratio and aliphatic carbon ratio is 1, so $f_{\mathrm{al}}^{\prime }$ was calculated according to $f_{a}$. Δ$f_{\mathrm{al}}$ is the difference between the aliphatic carbon ratio calculated by $f_{\mathrm{al}}^{H}$, $f_{\mathrm{al}}^{*},$ and $f_{\mathrm{al}}^{O}$ and $f_{a}$. From Δ$f_{\mathrm{al}}$, it can be seen that the aliphatic carbon ratios obtained by the two calculation methods are extremely close, indicating that the NMR spectrum fitting calculation results are relatively accurate.

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
