# Peer review of "Fine Characterization of the Macromolecular Structure of Huainan Coal Using XRD, FTIR, 13C-CP/MAS NMR, SEM, and AFM Techniques"

_molecules, 2020, doi:10.3390/molecules25112661_

Round 1

Reviewer 1 Report

The present manuscript is acceptable.

Author Response

Thank you very much for your constructive comments on our work. On the basis of fully studying the relevant opinions, we have carefully revised the article and obtained your approval. Here we express our most heartfelt thanks for your support. For our follow-up research results, please continue to pay attention and give us guidance.

Reviewer 2 Report

 In this version the authors have made significant improvements of their original manuscript, They also answered carefully the different points addressed. It can be considered worthwhile for publication in Molecules.

Just a few remaining comments and  corrections:

It might be valuable to clearly give, in the introductory section, the aim of the structural analyses of this Huainan coal sample which is to develop in the near future graphene based on Huainan coal.

I think that the sentence page 3 line 95  “ When the magnification is increased, the fine structure of a single macromolecule can be observed” should be remove since it appears very speculative  anything…

Page 14 Figure 8 the caption on the horizontal scale dC/% should be replace by 13C  ppm

Again be very careful with the quality of the different Figures

Author Response

First of all, I would like to express our sincere thanks and offer valuable and constructive opinions on our work, which has played an important role in improving our work level.
Secondly, we have made item-by-item amendments in strict accordance with your requirements.
1. The introduction adds the value of this study to the subsequent preparation of Huainan coal-based graphene.
2. The expression was deleted.
3. The X-axis label in Figure 8 has been modified.
4. We have improved the quality of the picture.
Finally, please continue to guide us on our subsequent research results.

This manuscript is a resubmission of an earlier submission. The following is a list of the peer review reports and author responses from that submission.

Round 1

Reviewer 1 Report

In this paper, the authors describe the characterization of macromolecular structure of Huainan coal by using some instruments.

This is a nice work, which is benefited to estimate of Huainan coal. Although this paper may be of value for publication in the field of coal chemistry, in order to ensure the publication of the current manuscript, I would like to recommend the authors to add the structural features of obtained Coal by SEM or TEM images for explaining the macro molecular structure.

Please revise the figure graphics as the text in the graphic is not legible/will not be legible in publication

Reviewer 2 Report

Fine characterization of macromolecular structure of Huainan coal using XRD, FTIR and 1a3CCP/MAS NMR

Dun Wu, Hui Zhang, Guangqing Hu, Wenyong Zhang

In this work the authors using classical CPMAS 13C NMR complemented by FTIR, XRD and ultimate analysis investigated the macromolecular structural properties of a coal samples from Huainan coalfield.

The overall work is rather classical in terms of NMR and combined techniques. It is just an illustration that these techniques used together lead to a better characterization of the macromolecular structure of coal samples. Nevertheless, it is difficult to see the real inputs of this work compared with all the literature already published in this field.

Although the introduction section is well documented and interesting, I can’t recommend this work to be published in Molecules

the following points should be corrected or improved:

The analyses seems to be performed on only one sample (this point is not clear). More samples should be analyzed to have a statistic representative of the whole field coal.
The authors claimed that is important to used demineralized samples. The data carried out on crude samples should be also produced to see the influence of such treatment on the obtained structures.

Some figures are of poor quality on the provided manuscript and should be improved.

Row (NMR, FTIR…) spectroscopic data should be given at least as supplementary information;

The sentence “Sullivan and Macielge used dipole phase shifting and cross polarization/magic angle rotation (CP/MAS) 13C NMR techniques” contains several mistakes:

Maielge should replace by Maciel.

I am not sure that “dipole phase shifting” is a existing sequence

The beginning of the sentence “The application of spectral analysis technology to characterize…” is not clear

“fitted by PeakFit software, and” the editor of PeakFit Software should be given

Please check the quality of Table 2 and the propose structural model is very difficult to read

Figure 6 13C NMR spectra are presented in a non conventional way. 0 ppm should to the right

“The sentence when the delay time t=40 μs” is not clear since no NMR sequence is precises.

The calculation of the the different structural parmeters (f ) form cP/MAS data is questionable in the present work since it does seem that the data were recorded under quantitative conditions : Optimum contact times is different for aromatic and aliphatic carbons for instance, moreover the presence of spinning side bands is expected for aromatic carbons and should be taken into account for quantitative analyses.